# Differences in the Sulfate–Methane Transitional Zone in Coastal Pockmarks in Various Sedimentary Environments

Chao Cao [1,2,3,4,*], Feng Cai [1,2,4], Hongshuai Qi [1,2,3,4], Shaohua Zhao [1] and Chengqiang Wu [1]

1   Third Institute of Oceanography, Ministry of Natural Resources, Xiamen 361005, China; caifeng@tio.org.cn (F.C.); qihongshuai@tio.org.cn (H.Q.); zsh20201987@163.com (S.Z.); wuchengqiang@tio.org.cn (C.W.)
2   Fujian Provincial Key Laboratory of Marine Ecological Conservation and Restoration, Xiamen 361005, China
3   Fujian Provincial Station for Field Observation and Research of Island and Costal Zone in Zhangzhou, Xiamen 361005, China
4   Southern Marine Science and Engineering Guangdong Laboratory (Zhuhai), Zhuhai 591000, China
*   Correspondence: caochao@tio.org.cn; Tel.: +86-592-2195306

**Abstract:** Different types of pockmarks, including single pockmarks, circular pockmarks, elongated pockmarks, chain-type pockmarks, and compound pockmarks, were identified in coastal areas around Fujian, China. The sediments associated with pockmarks were mainly silty clay to clay, with a small quantity of silt with fine sand. The sulfate content in the pore water in the sedimentary layers associated with pockmarks decreased with depth from the surface, whereas the free methane content increased with depth. The interaction between sulfate and methane is well known, but differences in the sulfate–methane transitional zone (SMTZ) were observed in different areas with different hydrologic characteristics. The sedimentary SMTZ of the offshore Zhe-Min mud wedge was shallow, at 50–70 cm below the seafloor. The sedimentary SMTZ was moderately deep (90–115 cm) in the central bay area and deep (180–200 cm) in the sandy area offshore. This variability in SMTZ depth reflects different amounts of free methane gas in the underlying formations, with a shallower SMTZ indicating a higher free methane content. The free methane had $\delta13C$ values of $-26.47‰$ to $-8.20‰$ and a biogenic hybrid genetic type. The flux of sedimentary gas from the pockmark surfaces, calculated according to Fick's formula, was 2.89 to 18.85 L/m$^2$·a. The shape, size, and scale of the pockmarks are directly related to the substrate type and the gas production of the underlying strata and thus vary with the sedimentary environment and development stage. Therefore, different types of pockmarks, in various phases of development, are associated with different sedimentary and dynamical conditions. A single circular pockmark is formed by a strong methane flux. As the intensity of methane flux weakens, the pockmark becomes elongated in the direction of the water flow because of long-term erosion induced by regular hydrodynamic forces. Finally, under a weak intensity of methane flux and the influence of complex hydrodynamic conditions, pockmarks merge to form large-scale compound pockmarks.

**Keywords:** pockmark; sulfate and methane transitional zone; various sedimentary environments; pore water; sediment

## 1. Introduction

Submarine pockmarks were first discovered in 1970 during a submarine oil and gas exploration process on the continental shelf of Nova Scotia in Canada [1]. However, these crater-shaped submarine depressions did not become widely known. It was not until 1987 that Hovland reported features described as hemp pits in North Sea sediments in authigenic carbonate cement and confirmed that pockmark formation is related to methane leakage events. Researchers gradually realized that pockmarks may indicate past and present submarine fluid activity. With the development of seismic exploration technology and marine acoustic detection technology, an increasing number of submarine pockmarks have

been found globally, such as on the continental slope of northern Norway, the continental slope of equatorial West Africa, the Bering Sea, the North Sea, the continental shelf of western Canada, the Gulf of Mexico, the Black Sea, the East China Sea, and the South China Sea [2–9]. Due to complexities and differences in marine environments and submarine fluid activities, pockmarks show different sizes, morphological characteristics, distribution locations, and water depths. Moreover, the biogeochemistry of pockmark sediments can indicate the formation environment of the pockmarks [10].

It is well known that a large quantity of sulfate is a major component of sedimentary pore waters in the early stages of sediment formation in estuarine and offshore areas [11–15]. Through the decomposition of sedimentary organic matter and the resulting increase in methane concentration, pore water sulfate concentrations decrease over time [16–19]. This can also lead to abnormal chloride concentrations and oxygen and hydrogen isotope values, and these anomalous values can be used to differentiate various sedimentary environments [20–23]. A sharp linear reduction in pore water sulfate concentrations and a relatively shallow sulfate–methane transitional zone (SMTZ) reflect a high intensity of methane flux through the underlying sedimentary layers and suggest an anoxic sedimentary environment [8,24,25]. Thus, sulfate and halide concentrations, a high intensity of methane flux, authigenic carbonate minerals, and other geochemical indicators are significant factors in studies on early diagenesis and the sedimentary environment [26–28]. Estuaries and coasts are the convergence zones of the land and sea, into which large amounts of terrestrial organic matter are input every year [29]. Higher primary productivity and deposition rates are promoted through the mixing of saline and fresh water. Thus, estuarine and coastal sediments tend to be rich in organic matter [30–32]. This organic matter can provide energy and electron donors for sulfate reduction, the intensity of methane flux, and other biogeochemical processes [33–35]. The intensity of methane flux and sulfate reduction are the main processes of the biogeochemical cycle in estuarine and coastal sediments and are also the foundation of the sedimentary carbon and sulfur biogeochemical cycles [36–38]. The sediment surface in various environments shows pockmarks with different shapes, sizes, and formation mechanisms as a result of the combined action of hydrodynamic forces, sediment transport, and shallow subsurface gas flow [39,40]. The biogeochemical processes and sedimentary environments of such pockmarks are areas of increasing interest to researchers [41,42].

The region offshore of Zhejiang and Fujian Provinces represents the largest underwater bank slope-type argillaceous deposit in China's coastal zone [43]. Due to its wide distribution range, large deposition thickness, high source diversity, complex hydrodynamic conditions, and other factors, this area provides an excellent natural test site for the study of Quaternary environmental evolution [44,45]. Since the late Pleistocene, the Taiwan Strait has been affected by tectonic compression, extension, and compression again, along with sea-level fluctuations and the uneven uplift and subsidence of local blocks, resulting in a complex submarine topography and a complicated sedimentary environment. Early investigations showed that the methane concentration in the shallow sedimentary environment was relatively high, so it was speculated that there might be abnormal characteristics such as oil and gas source rocks in the underlying strata. In recent years, both the special investigations of Comprehensive Survey and Evaluation of China's Offshore Marine Resources and geological support engineering have found a large number of pockmarks and shallow gas outflow in this area, and they appear in areas with different types of deposits [46]. Although pockmark landforms are common on the seafloor of continental margins globally, little is known about the geochemical characteristics of the sediment, activity status, trigger mechanism, and formation time of submarine pockmark landforms, especially shallow-water landforms. Previous research shows that most pockmarks are inactive, but what is the current status of shallow-water pockmarks? Is it possible to obtain some information on the activity of pockmarks through pore water geochemistry? The formation of pockmarks is usually related to the leakage of submarine fluids. Therefore, what are the main biogeochemical reaction rates in sediments and the contributions of ac-

tive fluids, such as methane and dissolved inorganic carbon, to shallow-water pockmarks? Understanding these processes is also important to understanding the global carbon cycle. Therefore, in this study, we selected pockmark landforms with different sedimentary environments in the coastal waters of Fujian Province to determine the spatial characteristics of the SMTZ distribution, total organic carbon (TOC) content, dissolved inorganic carbon (DIC) content, and $\delta^{13}C$ values in pockmark landforms with different morphologies, as well as other important indicators of sedimentary environment types, combined with the characteristics of stratigraphic structure, hydrodynamic conditions, and regional material sources. This paper analyzes the genetic mechanism and development process of different forms of pockmark landforms and preliminarily constructs a development model of pockmark landforms in different sedimentary environments.

## 2. Study Area

The offshore area of Fujian ($32°33'$–$27°11'$ N and $117°10'$–$120°50'$ E) in the western Pacific Ocean is adjacent to southeastern China and is bounded by the Taiwan Strait to the east, Zhejiang to the north, and Guangdong to the south (Figure 1) [47]. The main sediment sources of coastal accumulation landforms are sediments carried by runoff in mountain streams and rivers. These sediments are transported to the coast by runoff or directly washed into the alongshore current system, and their input affects the erosional and depositional processes of beaches [40,46]. The total watershed area in Fujian is $11.03 \times 10^4$ km², and the average annual total amount of outflow water is $10{,}380.54 \times 10^8$ m³ (1956–2018). The Minjiang River accounts for 55.4% of the total, the Minnan rivers supply $2910 \times 10^8$ m³ (28% of the total), and the Mindong rivers supply $1720 \times 10^8$ m³ (16.4% of the total). The average annual amount of sediment entering the sea in the whole province is $2000 \times 10^4$ t, and the Minjiang, Jiulongjiang, and Jinjiang Rivers account for 50% of the total sediment load.

The coastal geology of Fujian is part of the western Pacific trench–island arc–marginal sea system and is affected by the action of four plates: Eurasia, India, Pacific, and Philippine Sea [49]. The Quaternary neotectonic activity is characterized by the differential uplift of fault blocks and by fault activity, and the main tectonic deformation is the reactivation of NE (northeast), NNE (north-northeast), NW (northwest), and NNW (north-northwest) trending faults that originally formed in the Yanshanian. Based on the linear features generated by the NW–SE and E–W trending faults, the intensity of faulting since the late Pleistocene increased seaward from the land to the ocean. The differential activity of the faults and fault blocks has controlled the development of the Quaternary basin, plain, and bay. In the coastal areas, the activity of the Changle–Zhaoan fault zone has controlled the distribution of the Quaternary basin and plain in the northeast along with the development of an NE trending graben and horst system and a bay. The fault activity in the northwest has been greater since the late Pleistocene and has controlled most of the basin development in this area. The offshore area is to the west of a fault in the littoral zone, and intense fault-related subsidence has prevailed in the Taiwan Strait during the Quaternary. Consequently, numerous fractures and vertical conduits have developed in the coastal sediments due to crisscrossing tectonic structures, providing important passages for subsurface gas movement [50,51].

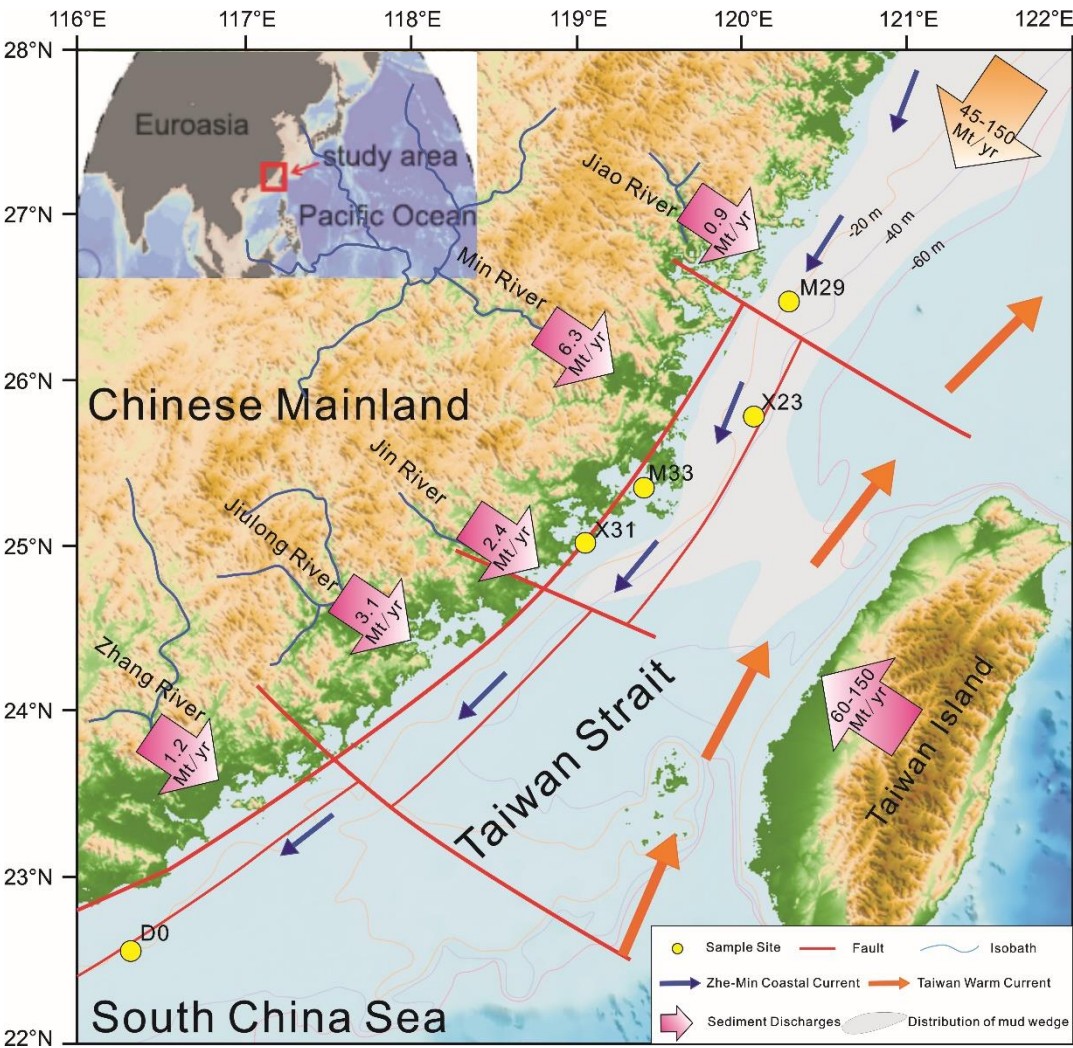

**Figure 1.** Location map and sampling stations of the study area. In this map, isobaths of 20, 40, and 60 m are shown by the thin light-yellow, purple, and red lines, respectively. Blue and orange arrows indicate the Zhe-Min Coastal Current and Taiwan Warm Current, respectively [48]. Arrows marked with gradual magenta and numbers indicate fluvial sediment discharge (Mt/year) from the Chinese mainland [29] and Taiwan. The mud wedge on the inner shelf of the East China Sea is marked with translucent gray areas. Fluvial drainage systems surrounding the TS (total sediment) and location of cores M29, X23, M33, X31, and D0 (yellow circle).

## 3. Materials and Methods

### 3.1. Field Work

#### 3.1.1. Bathymetric Measurements

From May 2014 to September 2016, a multibeam echosounder (SeaBeam 3012, L3, Kiel, Bavaria, Germany) was used to investigate the submarine topography of muddy areas offshore of northern Fujian, estuarine areas off central Fujian, sandy areas offshore of southern Fujian, and other coastal areas. When the water depth was less than 10 m, the distance between survey lines was set at 50 m. In areas where the water depth was 10–20 m, the distance between survey lines was 100 m, and when water depths were >20 m, the distance between the survey lines was 200 m. The distance between the survey lines was changed according to the actual water depth in the survey area, the sweep width of the instrument, and sea conditions. An overlap of 10% between adjacent survey amplitudes was ensured, and a total area of 500 km$^2$ was surveyed with the multibeam echosounder.

### 3.1.2. Sediment Collection

In May 2014 and June 2015, a scientific research cruise on the ships *Runjiang* and *Yanping II* recovered sediment core samples from five locations in the nearshore area of the western Taiwan Strait (Table 1, Figure 1). The five sites are located in the muddy areas off Zhejiang and Fujian (M29 and X23), in Xinghua and Meizhou bays offshore of central Fujian (M33 and X31), and in the sandy area offshore of southern Fujian (D0). The water depths were 15–60 m, and the core lengths were 106–153 cm. Samples from M33 and X31 had high sedimentary water and organic matter contents and an obvious putrid odor and were mainly composed of black-gray clay with plant debris. Samples from M29 and X23 were high in water content, and the organic matter decreased with distance from land; the sediment had a slightly fishy odor and was composed of grayish-green sandy clay containing little biodetritus and plant root residues. Samples from D0 had low water and organic matter contents and were composed of yellowish-green silty sand with little biodetritus. The bottom water temperatures were 26.65–29.64 °C, and the salinity was 32.11–33.97.

**Table 1.** Sampling locations and parameters measured in the field at the five stations.

| Stations | Longitude | Latitude | Water Depth (m) | Sample Length (cm) | Temperature (°C) | Salinity | Sediment Type |
|----------|-----------|----------|-----------------|--------------------|------------------|----------|---------------|
| M29 | 120°20′18″ | 26°28′10″ | 26 | 146 | 26.65 | 33.66 | Clay |
| X23 | 120°08′25″ | 25°42′40″ | 38 | 153 | 27.07 | 33.97 | Clay |
| M33 | 119°25′18″ | 25°20′18″ | 16 | 135 | 29.17 | 32.11 | Silt |
| X31 | 119°00′01″ | 25°3′59″ | 19 | 144 | 28.33 | 32.82 | Silt |
| D0 | 116°10′17″ | 22°41′59″ | 22 | 106 | 29.64 | 33.21 | Sand |

Sediment samples from the five locations were recovered with a gravity pipe sampler (counterweight = 1 ton; length = 4 m; inner diameter = 75 mm). Holes were drilled in the outer wall of the PVC casing at 5 cm intervals ($\varphi = 1$ cm) and then sealed with electrical adhesive tape before sampling. The pipe wall was wrapped with a film and sealed with the same adhesive tape. The sediment cores were arranged horizontally on an indoor bearing platform, and 3–5 cm samples were taken and placed into bottles (50 cm³, Bellco, Vineland, NJ, USA) under anaerobic conditions while inhibiting methanogens added in advance (1 moL/dm³ NaOH). The bottle openings were sealed with a rubber plug after nitrogen purging, and the bottles were sealed with aluminum caps, labeled, and stored cold. The sedimentary headspace methane content and carbon isotopes were measured using these samples. In addition, Macro Rhizon pore water samplers (sampling head of 5 cm) were inserted into the sediment core in the premanufactured holes and were equipped with 5 cm³ vacuum tubes. Sediment pore water was extracted under negative pressure on site; 2–3 tubes of pore water were collected from each layer, sealed with Parafilm (Sigma-Aldrich, St. Louis, MO, USA), labeled, and preserved in cold storage. The cores were cut into 1 m sections upon completion of pore water extraction and preserved in a refrigerated locker (4 °C) after being wrapped and sealed.

### 3.2. Data Processes

#### 3.2.1. Bathymetric Data Process

The original multibeam data were processed by sound velocity profile corrections, tidal level corrections, and parameter corrections; noisy points were eliminated effectively. Finally, a high-precision submarine digital elevation model (grid resolution of 50 m) was established, and the seabed slope and terrain profile were calculated to analyze the relief changes in the study area and to determine the size, shape, distribution, and other related parameters of the pockmarks.

#### 3.2.2. Laboratory Analysis of Sediment Samples

The analysis was performed with an Iso prime 100 continuous flow isotope ratio mass spectrometer (CF-IRMS, Finnigan, San Francisco, CA, USA). For each sample, 0.5 mL of

sample was placed into a glass bottle with a round bottom, high-purity helium was blown over the sample, and 5–6 drops of anhydrous phosphoric acid were added. After reaction at 40 °C for 4 h, the produced $CO_2$ was separated by a chromatographic column and then sent to a mass spectrometer for carbon isotope determination. The analysis accuracy was better than 0.2‰. The linear formula relationship between the signal intensity of $^{44}CO_2$ and the DIC (dissolved inorganic carbon) contents of 0.85, 2.07, and 3.93 mmol/L DIC samples was established to calculate the DIC content. Each sample was measured twice, the experimental error was less than 5%, and the standard deviation was $\pm 0.2\%$. The reference standard for carbon isotopes is V-PDB (the reported carbon isotope values are the corrected result based on the international standard V-PDB). The DIC contents and $\delta^{13}$C-DIC values were analyzed on the science and technology sharing experimental platform of the Third Institute of Oceanography, Ministry of Natural Resources.

The pore water was diluted 500 times with ultrapure water. The anions $Cl^-$ and $SO_4^{2-}$ and cations $Na^+$, $K^+$, $Mg^{2+}$, and $Ca^{2+}$ were measured by the Diane company's Dionex ICS-900 ion chromatograph. When analyzing anions, 4.5 mmol $Na_2CO_3$ and 0.8 mmol $NaHCO_3$ were used as the eluent, 25 mmol $H_2SO_4$ as the regeneration solution, 50 μL as the injection volume, and 1 mL/min as the flow velocity, and the chromatographic column was an IonPAC AS19. Repeated measurements of standard seawater showed that the standard deviations of all anions and cations were less than 0.5% [52]. The pore water pH was measured with a CHN 86801 (Orion, Boston, MA, USA) pH meter (accuracy of $\pm 0.001$). The ion content of the pore water was analyzed on the science and technology sharing experimental platform of the Third Institute of Oceanography, Ministry of Natural Resources.

An amount of 3 mL of sediment in the weighing bottle was dried at 105 °C and weighed after achieving a constant weight [53]. The weights of the sample before and after the experiment were then compared. The porosity was presented by the volume of water contained in the sediment per unit volume. The sediment samples were pretreated by removing impurities, grinding, and acidification. The sediment samples were pretreated to remove impurities, ground, and acidified, after which a Vario EL-III (Elementar, Nottingham, UK) elemental analyzer was used to measure the TOC and total nitrogen (TN), and a Delta Plus II XL isotope ratio mass spectrometer (IRMS) was used to determine the $\delta^{13}C_{TOC}$. The standard deviation (σ) was $\pm 0.2\%$ (IRMS). The sediment samples were analyzed at the State Key Laboratory of Marine Geology (Tongji University, Shanghai, China).

3.2.3. Methane Flux Calculation

Sedimentary organic matter can affect the diffusion of $CH_4$; thus, the $CH_4$ diffusion flux in sediments in the study area was calculated according to Fick's laws (Equations (1) and (2)) [54]:

$$J = -\varphi Ds \frac{dc}{dx},\tag{1}$$

where $J$ is the diffusion rate (mmol/(m$^2$·a)), $\varphi$ is the sediment porosity, $Ds$ is the sediment diffusion coefficient (m$^2$/a), $c$ is the $CH_4$ concentration (mmol/dm$^3$), and $x$ is the sediment depth (m).

$$Ds = \frac{D_0}{1 + n(1 - \varphi)},\tag{2}$$

where $n = 3$ (lithology factor of silty clay) and $D_0 = 1.4 \times 10^{-5}$ cm$^2$/s (initial diffusion coefficient of methane at 20 °C) [54].

## 4. Results

### 4.1. Morphological Features of Various Pockmarks

According to the multibeam data, several types of pockmarks were identified. The main types included (1) small pockmarks, which have widths of 1–10 m, diameters of <5 m, depths of up to 0.5 m, and a wide distribution throughout the study area; (2) circular pockmarks (diameter = 5–30 m, depth = 1–5 m), which have bowl-like cross sections (inner

slopes of the pockmark are gentle) or asymmetric cross sections (inner slopes are steep), in which single pockmarks are widely distributed around these more normal pockmarks; (3) elongated pockmarks (depth = 1–8 m), which have a much longer axis and often appear on slopes or areas affected by strong bottom water flow and mainly distributed in the northern muddy areas; (4) chain-type pockmarks, which are small pockmarks or small normal pockmarks arranged in a line or curved shape and have lengths that can extend for thousands of meters because they are generally formed from concentrated fluid leakage along approximately vertical fault or weak zones, and are mainly found in estuaries and bays; and (5) compound pockmarks, which are normal pockmarks appearing in groups or formed through a combination of several pockmarks, and are mainly distributed in the southern sandy area (Table 2).

**Table 2.** Distribution and types of pockmarks in the study area.

| No. | Area | Types | Characteristics | Sedimentary Type |
|---|---|---|---|---|
| 1 | Zhe-Min muddy | Unit, circular, liner | Medium to small diameter, depth of pockmark not exceeding tens of meters, linear distribution, the overall size not large | Silty clay |
| 2 | Bay mouth | Unit, circular, elongated | Medium to small diameter, depth of pockmark not more than a few meters, shape asymmetry, the overall size small or medium | Clayey silty |
| 3 | Sandy | Circular, elongated, mixed | Medium to large diameter, depth of pockmark not more than 10 m, asymmetrical form, appearing in groups, large scale | Silty sand |

### 4.2. Geochemical Features of Pore Water in the Sediment

4.2.1. $SO_4^{2-}$ and $CH_4$ Concentrations in Pore Water

The $SO_4^{2-}$ and $CH_4$ concentrations in sediment pore water from five stations (M29, X23, M33, X31, and D0) are shown in Figure 2. The $CH_4$ concentrations in pore water gradually increased with sediment depth, while $SO_4^{2-}$ concentrations gradually decreased. The $SO_4^{2-}$ concentrations in the surface sediment pore water were higher, and the $SO_4^{2-}$ concentration profiles showed a linear decrease with depth; the correlation coefficients ($r$) were 0.83 (M29), 0.92 (X23), 0.75 (M33), 0.71 (X31), and 0.55 (D0) until the $SO_4^{2-}$ was consumed completely. The corresponding sediment depths where the pore water $SO_4^{2-}$ was consumed completely were 65 cm (M29), 55 cm (X23), 95 cm (M33), 110 cm (X31), and 200 cm (D0); these depths corresponded to the SMTZ. The surface sediments at station D0 were silty sediments containing incompletely decomposed humus. Consequently, the microbial degradation was not complete, and the active organic matter content was low. Thus, the geochemical activity in the surface sediment pore water at D0 was not obvious.

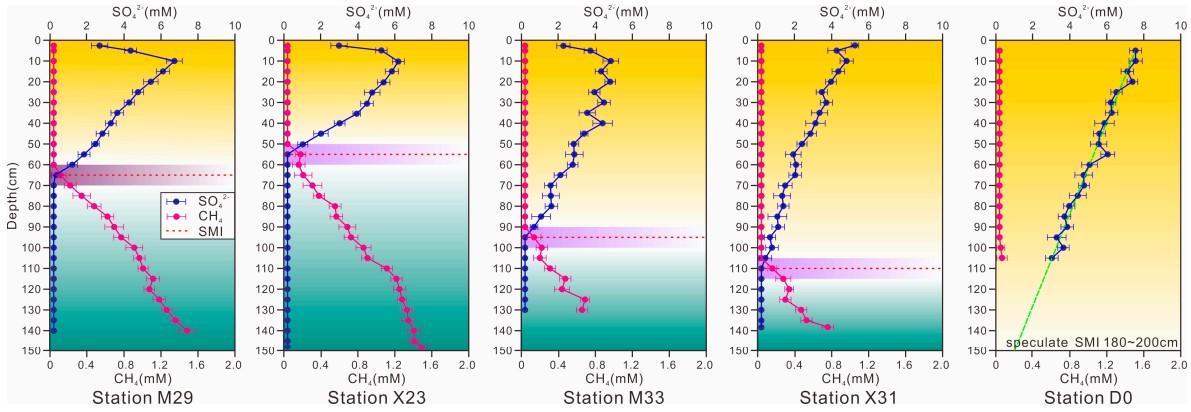

**Figure 2.** Concentration profiles of sulfate and methane in pore water.

The changes in pore water $CH_4$ and $SO_4^{2-}$ concentrations were obviously different. The $CH_4$ concentrations in the pore water were relatively low in the upper sedimentary layers and exhibited only slight variations, but the $CH_4$ concentrations sharply increased in the vicinity of the SMTZ. The $CH_4$ concentrations rapidly increased from 0.02 to 1.57 mM at 140 cm at M29, 0.04 to 1.58 mM at 150 cm at X23, 0.03 to 0.70 mM at 125 cm at M33, and 0.04 to 0.78 mM at 140 cm at X31. However, limited $CH_4$ was found at D0, with only a minor concentration at 100 cm.

### 4.2.2. DIC Concentrations and $\delta^{13}C_{DIC}$ in Pore Water

The DIC concentrations and $\delta^{13}C_{DIC}$ values of the sediment pore water are shown in Figure 3. The DIC concentrations of the sediment pore water gradually increased with depth at all sites until the maximum values were reached in the vicinity of the SMTZ. The concentrations then decreased and generally remained stable; however, the pore water DIC increased again at the bottom of some stations. At M29, the pore water DIC in the surface sediment was 3.25 mmol/dm$^3$, increased to 12.08 mmol/dm$^3$ at 60 cm, and then gradually decreased with depth until 120 cm, below which it remained stable. At X23, the DIC increased from 4.02 to 16.24 mmol/dm$^3$ at 55 cm and then gradually decreased with depth, but a second concentration peak (11.89 mmol/dm$^3$) was present at 110 cm. At M33, the surface sedimentary DIC in pore water was 2.11 mmol/dm$^3$, increased to 10.42 mol/dm$^3$ at 100 cm, and gradually decreased with depth to a stable value at 120 cm. At X31, the DIC increased from 3.74 to 8.16 mmol/dm$^3$ at 115 cm and then gradually decreased with depth to a stable value. At D0, the DIC gradually decreased to 16.25 mmol/dm$^3$ from 0 to 105 cm.

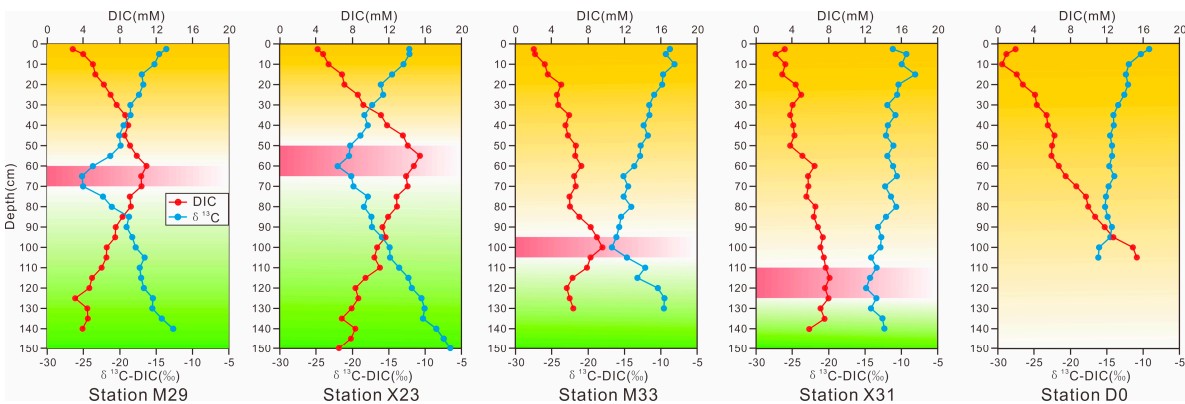

**Figure 3.** Concentration profiles of DIC and $\delta^{13}C_{DIC}$ in pore water.

The sedimentary pore water $\delta^{13}C_{DIC}$ values at M29, X23, M33, and X31 showed rightward convex curves with peak negative carbon isotope values of −25.43‰(M29), −22.91‰(X23), −17.75‰(M33), and −15.02‰(X31) at 65 cm (M29), 60 cm (X23), 100 cm (M33), and 120 cm (X31) below the sediment surface. The most negative $\delta^{13}C_{DIC}$ values were close to the SMTZ at all analyzed stations, except for D0. In general, the $\delta^{13}C_{DIC}$ values decreased with increasing depth to the SMTZ, below which they increased, except for D0.

The vertical changes in sedimentary TOC are shown in Figure 4. The apparent trends included the following: (1) In general, the surface sedimentary average contents of TOC gradually decreased from north to south at the sample station, with average contents of 1.78% (X23) >1.54% (M29) >1.22% (M33) >1.09% (X31) >0.74% (D0), and (2) sedimentary TOC gradually decreased with increasing depth. The TOC content was affected by the input of detrital terrigenous material. There was an obvious change in sedimentary TOC with depth at M29, but at sites farther south, there were smaller TOC changes from the sediment surface to the bottom of the column.

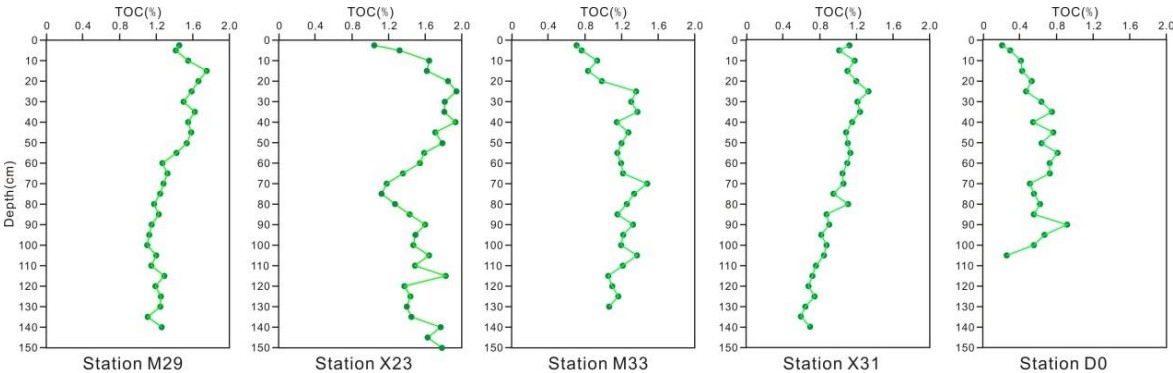

**Figure 4.** Component profiles of TOC in sediments.

### 4.2.3. Methane Flux

Sedimentary organic matter can affect the diffusion of $CH_4$. Thus, the $CH_4$ diffusion flux in the sediments of the study area was calculated according to Fick's laws (Equations (1) and (2)). The methane diffusion fluxes of the five stations, that is, M29, X23, M33, X31, and D0, were $15.17 \times 10^{-2}$ mmol·m$^{-2}$·a$^{-1}$, $18.85 \times 10^{-2}$ mmol·m$^{-2}$·a$^{-1}$, $9.31 \times 10^{-2}$ mmol·m$^{-2}$·a$^{-1}$, $8.72 \times 10^{-2}$ mmol·m$^{-2}$·a$^{-1}$, and $2.89 \times 10^{-2}$ mmol·m$^{-2}$·a$^{-1}$, respectively, with an average value of $10.99 \times 10^{-2}$ mmol·m$^{-2}$·a$^{-1}$ (Table 3). In combination with Figure 4, the TOC content distribution in sediments indicates that the input of organic matter in sediments is a key factor controlling the methane diffusion flux at the bottom. With an increase in the upward methane diffusion flux, the consumption of methane and sulfate, namely, the occurrence of the intensity of the methane flux process, is intensified and causes the upward migration of the SMTZ in the sediment.

**Table 3.** Sediment diffusion coefficients *Ds*, calculated diffusive intensity of methane flux, and the average content of TOC in surface sediments.

| Station | Porosity | *Ds* (m$^2$·a$^{-1}$) | *J* (mmol·m$^{-2}$·a$^{-1}$) | TOC (%) |
|---------|----------|------------------------|-------------------------------|---------|
| M29 | 0.65 | $2.53 \times 10^{-2}$ | $15.17 \times 10^{-2}$ | 1.54 |
| X23 | 0.71 | $2.75 \times 10^{-2}$ | $18.85 \times 10^{-2}$ | 1.78 |
| M33 | 0.80 | $3.29 \times 10^{-2}$ | $9.31 \times 10^{-2}$ | 1.22 |
| X31 | 0.76 | $2.65 \times 10^{-2}$ | $8.72 \times 10^{-2}$ | 1.09 |
| D0 | 0.85 | $4.04 \times 10^{-2}$ | $2.89 \times 10^{-2}$ | 0.74 |
| Average | 0.76 | $3.05 \times 10^{-2}$ | $10.99 \times 10^{-2}$ | 1.27 |

## 5. Discussion

### 5.1. The Formation Mechanism of Pockmarks

Pockmarks are craterlike depressions that appear in fine-grained sediments on the seafloor. They have received considerable attention because they can provide important information on fluid activities in continental margins [3]. Paul et al. (2008) believed that free gas first accumulates under fine-grained sediments with sealed capillaries [55]. As the amount of free gas increases, the gas buoyancy eventually exceeds the capillary pressure, and the free gas pierces the sealing layer and migrates into the overlying sedimentary layers, forming upward-flowing gas that entrains pore water as it discharges from the sediment. Due to the upward flow of gas and pore water, the surface sediments of the seabed are deformed, and the liquefied sediments are transported by the fluid flow, forming pockmarks. Consequently, fluids play an important role in the formation of pockmarks.

Because the seafloor stratigraphy and geomorphology are complex and changeable, pockmarks are affected by many factors after formation, such as continuous fluid leakage, bottom currents, tectonic activity, and sedimentation, with the result that the diameter, depth, and plan-view configuration of pockmarks can vary [10,56]. Weber et al. (2002)

proposed that the ratio of the slope and diameter of the inner wall to the depth of a pockmark reflects different development stages. The depth of newly generated or recently active pockmarks is deeper, and the inner walls are steeper, whereas pockmarks with larger diameters and shallower depths are older, as the bottoms have been infilled to some degree with sediment [57]. It can be seen from Figure 5 that most of the pockmarks in the study area are located in the offshore shallow sea area. Due to the thin sedimentary cover, strong hydrodynamic forces, and medium- and small-scale pockmarks, the depth and diameter of pockmarks are not more than 10 m. These pockmarks are mainly round pockmarks with shallow depths (1–2 m) and elongated pockmarks with greater depths (3–6 m). The boundary between individual pockmarks in the coarse particle sedimentary area is not obvious because of the strong hydrodynamic alteration following their formation.

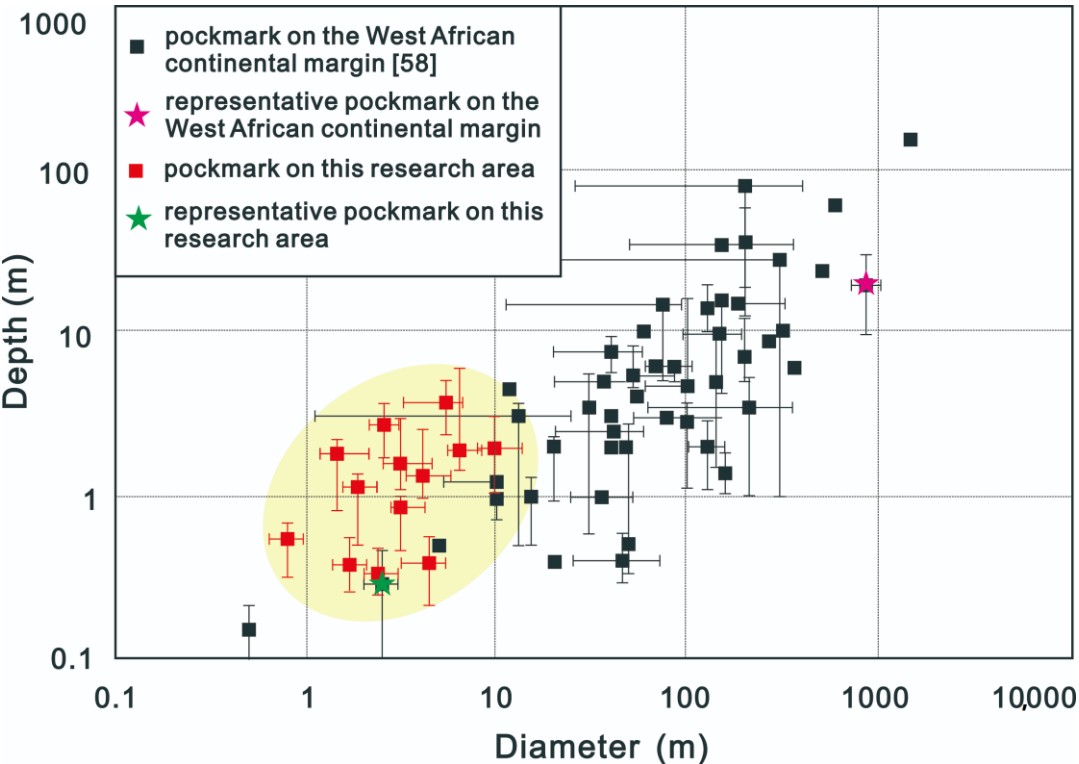

**Figure 5.** The relations diagram of pockmark diameter and depth. The line segment represents the variation range of the size of a single pockmark, and the point represents the size of a single pockmark area or the average value of the size of a single pockmark [58].

Hinrichs (1999) suggested that the formation of small pockmarks (with diameters <5 m) is related to the slow leakage of pore water, while the formation of more common pockmarks may be closely related to periodic or intermittent gas eruptions from the seabed [59]. Free gas in deeper strata can accumulate in shallow strata as bubbles with pore water filling the space between bubbles. With continuous gas accumulation, the pressure gradually increases until the storage strata rupture; the gas then erupts from the seabed along a fracture and forms pockmarks. Small pockmarks are mostly distributed along underwater slopes and plains of estuaries and coasts at water depths of 5–100 m. In these areas, the terrigenous input is abundant, and structural activity is common, which favors organic matter decomposition and gas accumulation in the sediments and the formation of pockmarks of different sizes on the seafloor [60–62]. As can be seen from Figures 6 and 7, the pockmarks in the estuary area (Figure 5, green asterisk) [8] feature shallow-water depths, small scales, and shallow SMTZs. The pockmarks in the submarine slope area (Figure 5, pink asterisk) [10,58] feature greater water depths, larger scales, and deep to nonexistent SMTZs. The pockmarks in the study area are present in a shallow-

water area, and the depths and scales of the pits are small. Affected by the sedimentary environment and hydrodynamic conditions, the morphologies and scales of the pockmarks are quite different.

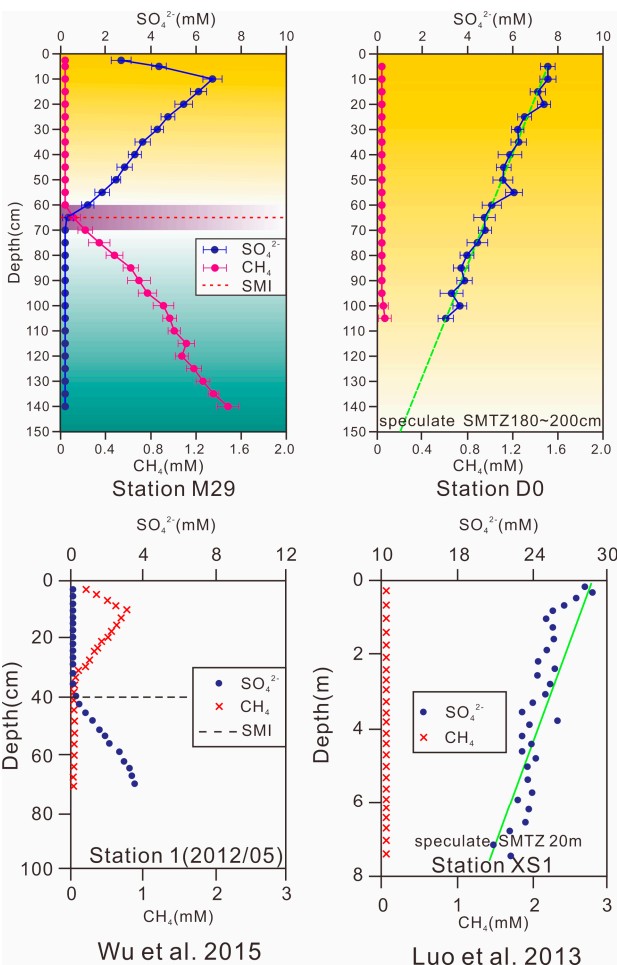

**Figure 6.** Depth comparison diagram of sulfate and methane transitional zone.

The results of physicochemical property and isotopic composition analyses of the shallow surface sediments associated with typical pockmark morphologies showed that (1) along the northern coast of Fujian, there is almost no input of terrestrially sourced materials from large rivers, the C/N ratio is indicative of a marine facies, and marine biogas is the main component based on the carbon isotopic composition. In the argillaceous zone, active materials are less abundant, and the sulfate content has a tendency to decrease, but this trend is not obvious. (2) Outside the mouth of the Minjiang River, in the underwater delta of the estuary, the substrate consists of sand and mud, the C/N ratio indicates the input of terrestrially sourced substances, active substances are relatively abundant, the reduction of sulfate is strong, the SMTZ is relatively shallow, and the carbon isotopic composition suggests that biogenic gas is common. (3) In the submarine depression offshore of southern Fujian Province (off Dongshan Peninsula), the carbon isotopic composition of the sandy bottom material shows that the organic matter is of mixed origin, originating from both the sea and land; the biogenic gas of mixed pyrolysis is derived from less active material (indicating that the underlying formation contains oil, gas, and methane); the sulfate reduction is strong; and the SMTZ is shallow.

### 5.2. Intensity of Methane Flux in Various Sediment Environments

The shapes and scales of various pockmarks are directly correlated with the sedimentary environment [61]. When the sediments feature a reducing environment, organic matter is oxidized by sulfate-reducing bacteria using pore water $SO_4^{2-}$ as an oxidizing agent in the reaction $2\,CH_2O + SO_4^{2-} \rightarrow 2\,HCO_3^- + H_2S$. Consequently, pore water $SO_4^{2-}$ concentrations decrease. Near the SMTZ, $SO_4^{2-}$ and $CH_4$ participate in the intensity of methane flux in the reaction $CH_4 + SO_4^{2-} \rightarrow HCO_3^- + HS^- + H_2O$; thus, sedimentary DIC mainly exists as $HCO_3^-$. As a result, pore water pH increases, equilibrium shifts to the right in the reaction, and $SO_4^{2-}$ also decreases with depth. With increasing $CH_4$ concentrations, the intensity of methane flux equilibrium moves to the right in the reaction equations, and $SO_4^{2-}$ decreases [62–64]. Borowski (1999) identified a strong intensity of methane flux in the upper stratum of a thick Quaternary sedimentary sequence on Blake Ridge [65], which explains the shallow SMTZ and rapid decrease in $SO_4^{2-}$ in the pore water profiles (Table 4). Thus, the depth of the SMTZ can indicate the strength of the intensity of methane flux in the sedimentary environment. The average SMTZ depth in the study area was approximately 180 cm. The SMTZ is shallower in the estuary than in other coastal areas because of the stronger intensity of methane flux.

**Table 4.** Compared SMTZ depths of pockmarks in the study area with other areas.

| Area | Water Depth (m) | SMTZ Depth (cm) |
|---|---|---|
| This study | 16~38 | 50~200 |
| Shenhu sea area | 800~1500 | 200~1750 |
| Baiyun depression | 1000~1200 | 600~1100 |
| Saanick bay | 225 | 10~20 |
| Skan bay | 10~65 | 20~30 |
| Black sea | 130~180 | 200 |
| Continental margin of Chile | 800~2700 | 220~400 |
| Caroline ridge and Blake submarine plateau | 1300~4750 | 1300~6000 |

In the early diagenetic process of marine shallow sediments, sulfate reduction is mediated by sulfate-reducing bacteria, which oxidize organic carbon, resulting in a change in the sulfate concentration gradient. Therefore, the oxidation of organic matter is the main reason for the change in sulfate concentration. In shallow marine sediments, the intensity of methane flux generally occurs in one sedimentary zone, and the upper transitional zone of this zone is known as the SMTZ [17,66]. The $CH_4$ generation zone is below the SMTZ. $CH_4$ is generated by reduction reactions between in situ microorganisms and $CO_2$ ($CO_2 + 4\,H_2 \rightarrow CH_4 + 2\,H_2O$) and acetate fermentation ($CH_3COOH \rightarrow CH_4 + CO_2$). $CH_4$ can also be generated through the decomposition of oil and natural gas hydrates [67,68]. The analysis of $CH_4$ offshore of Fujian showed that concentrations in shallow sediments from the estuaries increased with depth. Because the electron uptake capability of sulfate-reducing bacteria is relatively high, the activity of methanogenic bacteria is restricted; once most $SO_4^{2-}$ is reduced (more than 80%), $CH_4$ generation starts. Conditions that encourage a shallow SMTZ include a high deposition rate, rapid sedimentary burial, and high organic matter content. These conditions result in a strong reducing environment and increases the generation of $CH_4$, which produces a high intensity of methane flux (Table 3). Thus, the intensity of methane flux during diagenesis, particularly in the early stages of sedimentary burial, is closely related to the sediment deposition rate, organic matter content, porosity, and redox potential [69,70].

The methane concentration increases gradually below the SMTZ. The DIC produced by the intensity of methane flux inherits the relatively light carbon isotopic composition of methane, while the DIC produced by organic matter oxidation has a relatively heavy carbon isotopic composition. The diffusion coefficient of $^{12}CH_4$ is greater than that of $^{13}CH_4$ because of isotopic effects during diffusion, and $^{12}CH_4$ is oxidized first during the intensity of methane flux, increasing the $^{12}CH_4$ concentration gradient. As a result, the

diffusion of $^{12}CH_4$ into upper sediment layers increases. The amount of organic matter entering the deep $CH_4$ zone increases as the organic matter content of the surface sediments increases, and the $CH_4$ concentration and diffusion flux in sediment pore waters increase. The joint consumption of $CH_4$ and $SO_4^{2-}$ during the intensity of methane flux increases as the upward-diffusing intensity of methane flux increases and causes the SMTZ to shift into a shallower position in the sediment column [71,72]. As can be seen from Figure 4, the $\delta^{13}C$-DIC values of pore water at stations M29, X23, and M33 tend to decrease first and then increase with the change in depth, which is consistent with the change trend of the sulfate group and reaches the lowest value near the SMTZ, indicating that the intensity of methane flux has a strong effect. Although the high contents of organic carbon at stations X31 and D0 will also consume sulfate in pore water, there is no corresponding decrease in the organic carbon content of the layers with the sharp decrease in sulfate content in the sediments at these stations, indicating that the consumption of sulfate by organic carbon is not the main factor responsible for the sharp decline in sulfate ions in pore water.

### 5.3. Development Stages of Pockmarks and Application of This Information

The shape, size, and scale of pockmarks are directly related to the type of bottom material and the gas production flux of the underlying strata. Different sedimentary environments and hydrodynamic conditions produce different shapes, sizes, and scales of pockmarks, which indicate different development stages and different types of sedimentary environments. In the initial stage of pockmark formation, the pockmarks are mostly round or elliptic and are then subsequently affected by seabed collapse or bottom currents, forming various shapes, such as elongated and crescent shapes. Long-term fluid leakage is not a necessary condition for maintaining pockmark morphology [73,74]. Other mechanisms can also maintain and transform pockmark morphology. For example, a weak bottom current near the seafloor forms upwelling in pockmarks, which can prevent sediments from accumulating in pockmarks and thus maintain the original pockmark morphology [7,75].

Therefore, according to the shape, scale, substrate type, gas flux, and hydrodynamic conditions of the underlying strata in the study area, the following stages of pockmark geomorphology in the study area were identified. (1) In the initial stage of pockmark formation, individual pockmarks are usually round or oval in shape (Figure 7a), and the distribution of pockmarks is relatively independent. The intensity of methane flux plays a strong role in the sedimentary layer, fluid actively escapes the seabed, and multiple pockmarks can develop in lines, forming chain pockmarks. Such pockmarks are easily formed in the southern part of the Zhejiang and Fujian mud area. (2) Due to erosion by tidal currents and weak hydrodynamic forces, the pockmarks at the mouth of the bay are obviously elongated along the flow direction. The inner walls of the pockmarks gradually disappear over time due to erosion. The intensity of the methane flux effect weakens (Figure 7b), and circular and chain pockmarks gradually develop into elongated pockmarks. (3) For pockmarks developed in coarse sediments, due to the weak role of the intensity of methane flux (Figure 7c), the pockmarks that form in the initial stage are subjected to long-term complex hydrodynamic erosion and become integrated, resulting in the formation of complex pockmark morphologies with unclear types. Figure 7 shows the morphologic development patterns of the pockmarks in the study area.

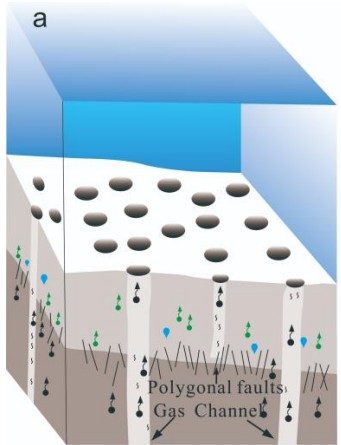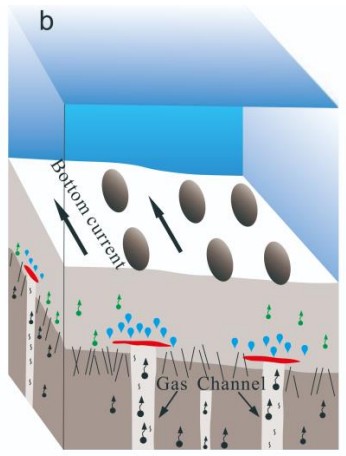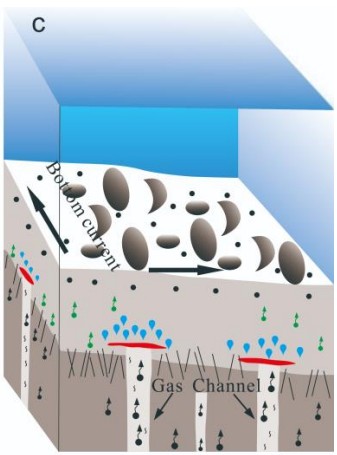

**Figure 7.** Pattern diagram of development stages and sedimentary environment of pockmarks. (**a**) Primary stage by the strong intensity of methane flux. (**b**) Growth stage by one-way flow. (**c**) Decline stage by the complex dynamic and weak intensity of methane flux.

## 6. Conclusions

The vertical profiles of pore water $CH_4$, $SO_4^{2-}$, DIC, and $\delta^{13}C_{DIC}$ values in areas offshore of Fujian Province showed that shallow SMTZs were the result of the strong intensity of methane flux activity in the study area. The pore water $SO_4^{2-}$ concentrations showed a linear decrease from the surface until the $SO_4^{2-}$ was almost completely consumed near the SMTZ. At this depth, $CH_4$ concentrations increased quickly, and the pore water DIC concentrations were obviously higher. The SMTZ depths at M29, X23, M33, X31, and D0 were 60, 55, 95, 110, and 200 cm, respectively.

The sedimentary organic matter and water (porosity) contents, along with microbial activity, affected the intensity of methane flux activity in the study area. Because the $^{12}$C oxidation rate is higher than that of $^{13}$C and $^{12}CH_4$ is heavier than $^{13}CH_4$, the $\delta^{13}C_{DIC}$ values become slightly more negative. Pore water $SO_4^{2-}$ was consumed through remineralization of organic matter, which increased the upward-diffusing $CH_4$ in the intensity of methane flux reaction zones; thus, the rate of consumed $SO_4^{2-}$ also increased correspondingly and shifted the SMTZ to a shallower depth. In addition, the sediment water content was high, and the active organic matter content increased; a portion of this organic matter was decomposed and consumed through mineralization, and consequently, the quantity of the active organic matter at the bottom of the sulfate reduction zone decreased. The $SO_4^{2-}$ in this zone reacted with $CH_4$ and was consumed, which promoted the intensity of methane flux.

The various development phases of pockmarks are characterized by different shapes, scales, and intensities of methane flux, which are in turn strongly correlated with different sedimentary environments. The single round pockmarks in the primary phase are normally associated with a strong intensity of methane flux. In contrast, elongated pockmarks are associated with a weakened intensity of methane flux and usually form under the control of stable bottom current flow, resulting in continuous erosion. Furthermore, compound pockmarks are characterized by a very weak intensity of methane flux and represent a relatively long-term adjustment to complicated hydrodynamic conditions.

**Author Contributions:** Designed the study, wrote the main manuscript, and prepared all figures: C.C.; contributed to the improvement of the manuscript: F.C. and H.Q.; collected the data: S.Z. and C.W. All authors have read and agreed to the published version of the manuscript.

**Funding:** This research was funded by the National Natural Science Foundation of China (Nos. 42076058, 41406059, and 41930538), the Scientific Research Foundation of the Third Institute of Oceanography of the Ministry of Natural Resources (No. 2019006), the Natural Science Foundation

of Fujian Province (No. 2016J01190), and the State Key Laboratory of Marine Geology at Tongji University (No. MGK1604).

**Institutional Review Board Statement:** Not applicable.

**Informed Consent Statement:** Not applicable.

**Data Availability Statement:** Data is contained within the article. The data presented in this study are available in Figure 2~Figure 4 and Table 1~Table 4.

**Acknowledgments:** We would like to thank the captain, officers, and crew of R/V *Yanping II* for their cooperation in collecting sediment cores in 2016. Chungen Liu and Yongbao Li from the Third Institute of Oceanography are thanked for their help in the sampling. The authors would like to express their sincere thanks to all those who have offered support.

**Conflicts of Interest:** The authors declare no conflict of interest.

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
