# Peer review of "Differences in the Sulfate–Methane Transitional Zone in Coastal Pockmarks in Various Sedimentary Environments"

_water, doi:10.3390/w13010068_

Round 1

Reviewer 1 Report

All comments/suggestions were made in the article itself.
The article is quite interesting showing the factors that led to the formation of the different types of holes found in the Taiwan Strait and showed that these are dynamic and that their development phases are characterized by different shapes, scales, and intensity of methane flux intensities, which are in turn strongly correlated with different sedimentary environments.

The work is well written and well structured and supported by figures and tables that are necessary for its understanding.
There are only minor issues to be resolved but that does not diminish the quality of the work.

Author Response

  1. L116: not black but blue colour. So, correct the colour of arrows or correct the word in the label of figure.

Response: We have modified black to blue arrows in the sentence. See this modification in line 117 of the manuscript with track changes.

  1. L278: according to the graph the initial value of DIC is not 4,02 but a lower value...

Response: We have modified value of DIC is 3.25 mmol/dm3 at M29. See this modification in line 282 of the manuscript with track changes.

  1. L282: correct the value according the graph because it is more or less 2 mmol and not 6,11 like you refer into the text...

Response: We have modified value of DIC is 2.115 mmol/dm3 at M33. See this modification in line 286 of the manuscript with track changes.

  1. L288: this pattern is not so evidente in stations X31 and D0, mainly in D0. In this last station i could say that the values decresed all over the vertical profile.

Response: We have modified the sentence “The sedimentary pore water δ13CDIC values at M29, X23, M33, X31 showed rightward convex curves with peak negative carbon isotope values of -25.43‰ (M29), -22.91‰ (X23), -17.75‰ (M33) and -15.02‰ (X31) at 65 cm (M29), 60 cm (X23), 100 cm (M33) and 120 cm (X31) below the sediment surface”. See this modification in line 293-296 of the manuscript with track changes.

  1. L292-293: “the δ13CDIC values decreased with increasing depth to the SMTZ, below which they increased.” This is not true for station D0.

Response: We have modified the sentence “In general, the δ13CDIC values decreased with increasing depth to the SMTZ, below which they increased, except for D0.” See this modification in line 297 and 298 of the manuscript with track changes.

  1. L295: From my analysis I notice that the stations M29 and X31 show a similar behavior of increase and then decrease (X31) or stabilize (M29). In the other stations, there is an increase up to a certain value and then there are oscillations around the maximum value. L295: What do you mean by north to south? Is it related to the position of the sampling sites in the study area? This needs to be clarified in the text so that there is no doubt.

Response: We have modified the sentence “In general, the surface sedimentary average contents of TOC gradually decreased from north to south at sample station, with average contents of 1.78% (X23) >1.54% (M29) >1.22% (M33) >1.09% (X31) >0.74% (D0)”. See this modification in line 300 and 301 of the manuscript with track changes.

  1. L343: The caption must be self-explanatory. Thus, there must be a reference to the red squares, green and pink asterisks, and the yellow area. In the case of the yellow area it may be perfectly identified in the graph itself ...

Response: We have added mean of red and black squares, green and red stars in the Figure 5. See this modification in line 347 of the manuscript with track changes.

  1. L364: Why these figures show the acronym SMI instead of SMTZ. The abbreviations used must be consistent throughout the text. Two abbreviations may not appear to mean the same, and SMI never appeared up to this point in the text ... so, the image must be replaced by placing the correct acronyms.

Response: We replaced SMI to SMTZ in the Figure 6 and checked all manuscript. See this modification in line 368 and 401 of the manuscript with track changes.

  1. L460: In the figure 7, replace the word Chinnel per channel.

Response: We replaced the word Chinnel to channel one by one in the Figure 7. See this modification in line 464 of the manuscript with track changes.

Reviewer 2 Report

Comments and Suggestions for Authors:

The presented paper can be published after minor revisions.

Small comments for the MS:

L83: What is “Mission 908»?

L187: DIC should be explained first time mentioned

L221: References are needed for the formula you use and the values of n and D0.

L247 Table 2. You put into “characteristics” column a phrase “depth does not exceed tens of meters”. Do you mean the depth of the pockmark or depth of the “Area” ?

L259-262 you write: “Consequently, the microbial 259 degradation was not complete, and the content of active organic matter was low. Thus, the 260 geochemical activity in the surface sediment pore water at D0 was not obvious, and the SO42− 261 concentration was low in the surface layer. 262

263», but the Fig.2 shows largest SO2 concentration in the surface sediment at D0 compared with other stations. Please check and correct.

L268. You use different unite in the text and in the Figure 2 (μmol/dm3 and mM),, please choose one.

L268: What is the precision of your technique, can you give results with 2 signs after comma (i.e. 1567.15 μmol/dm3)

L287: Fig.3. Please explain the colours belts meanings (red for SMTZ, right?)

L305: For what depth (x) you calculated “The methane diffusion fluxes” ? Was it the sediment water interface (SWI) or middle of SMTZ?

L343+345: Fig.5. What do you mean by red and black squares, green and red stars?

L364: What is SMI?

L387: “sedimentary DIC mainly exists as HCO3-“ . It happens not because of anaerobic methane oxidation, HCO3- dominates because of the sea carbonate system properties (typical alkalinity and typical total carbon content)

L396: References are needed for the locations you list in the table. Again, what is SMI? Is this the same as SMTZ?

Author Response

The presented paper can be published after minor revisions.

Small comments for the MS:

  1. L83: What is “Mission 908”?

Response: Full name of Mission 908 is Comprehensive Survey and Evaluation of China's Offshore Marine Resources. We modified Mission 908 to its full name. See this modification in line 83and 84 of the manuscript with track changes.

  1. L187: DIC should be explained first time mentioned

Response: Full name of DIC is dissolved inorganic carbon. We added its full name in line 199 at the first time mentioned.

  1. L221: References are needed for the formula you use and the values of n and D0.

Response: We added a reference NO.54 in order to explaining formula 1,2 and the values of n and D0. See this modification in line 226, 232 and 653 of the manuscript with track changes.

[54] Boudreau, B.P. Diagentic Models and Their Implementation: Modeling Transport and Reaction in Acquatic Sediments. Berlin Heidelberg: Springer Verlag, 1997, 414.

  1. L247 Table 2. You put into “characteristics” column a phrase “depth does not exceed tens of meters”. Do you mean the depth of the pockmark or depth of the “Area” ?

Response: The depth refers to the depth of pockmark. We modified depth to the depth of pockmark in Table 2. See this modification in line 250 of the manuscript with track changes.

  1. L259-262 you write: “Consequently, the microbial 259 degradation was not complete, and the content of active organic matter was low. Thus, the 260 geochemical activity in the surface sediment pore water at D0 was not obvious, and the SO42− 261 concentration was low in the surface layer. 262 263», but the Fig.2 shows largest SO2 concentration in the surface sediment at D0 compared with other stations. Please check and correct.

Response: First, We have deleted “and the SO42- concentration was low in the surface layer.” That description is wrong from the Figure 2. See this modification in line 264 of the manuscript with track changes.

  1. L268. You use different unite in the text and in the Figure 2 (μmol/dm3 and mM),, please choose one.

Response: We have confirmed unite of methane concentrations is mM.

  1. L268: What is the precision of your technique, can you give results with 2 signs after comma (i.e. 1567.15 μmol/dm3)

Response: According to the measurement accuracy of methane in this study. We have modified value and unite of methane, and keep two decimals at the value. Unite of methane concentrations is mM. See this modification in line 271~274 of the manuscript with track changes.

  1. L287: Fig.3. Please explain the colours belts meanings (red for SMTZ, right?)

Response: Yes. The red belts mean SMTZ.

  1. L305: For what depth (x) you calculated “The methane diffusion fluxes”? Was it the sediment water interface (SWI) or middle of SMTZ?

Response: The depth that we calculated the methane diffusion fluxes is the middle of SMTZ.

  1. L343+345: Fig.5. What do you mean by red and black squares, green and red stars?

Response: We have added mean of red and black squares, green and red stars in the Figure 5. See this modification in line 347 of the manuscript with track changes.

  1. L364: What is SMI?

Response: Full name of SMI is sulfate and methane interface. We modified SMI to SMTZ in the Figure 6. See this modification in line 368 of the manuscript with track changes.

  1. L387: “sedimentary DIC mainly exists as HCO3-“. It happens not because of anaerobic methane oxidation, HCO3- dominates because of the sea carbonate system properties (typical alkalinity and typical total carbon content)

Response: I quite agree with the reviewer's opinion that HCO3- dominates because of the sea carbonate system properties. But HCO3- dominates because of anaerobic methane oxidation at the SMZT.

  1. L396: References are needed for the locations you list in the table. Again, what is SMI? Is this the same as SMTZ?

Response: We added several references at the corresponding area in the table 4 and modified SMI to SMTZ in the table 4. See this modification in line 401 of the manuscript with track changes.
